# Distributionally Robust Optimisation with Bayesian Ambiguity Sets

**Charita Dellaporta**[*]
Department of Statistics
University of Warwick
C.Dellaporta@warwick.ac.uk

**Patrick O'Hara**[*]
Department of Computer Science
University of Warwick
Patrick.H.O-Hara@warwick.ac.uk

**Theodoros Damoulas**
Department of Computer Science & Department of Statistics
University of Warwick
T.Damoulas@warwick.ac.uk

## Abstract

Decision making under uncertainty is challenging since the data-generating process (DGP) is often unknown. Bayesian inference proceeds by estimating the DGP through posterior beliefs about the model's parameters. However, minimising the expected risk under these posterior beliefs can lead to sub-optimal decisions due to model uncertainty or limited, noisy observations. To address this, we introduce Distributionally Robust Optimisation with Bayesian Ambiguity Sets (DRO-BAS) which hedges against uncertainty in the model by optimising the worst-case risk over a posterior-informed ambiguity set. We show that our method admits a closed-form dual representation for many exponential family members and showcase its improved out-of-sample robustness against existing Bayesian DRO methodology in the Newsvendor problem.

## 1 Introduction

Decision-makers are regularly confronted with the problem of optimising an objective under uncertainty. Let $x \in \mathbb{R}^d$ be a decision-making variable that minimises a stochastic objective function $f : \mathbb{R}^d \times \Xi \to \mathbb{R}$, where $\Xi$ is the data space and let $\mathbb{P}^\star \in \mathcal{P}(\Xi)$ be the data-generating process (DGP) where $\mathcal{P}(\Xi)$ is the space of Borel distributions over $\Xi$. In practice, we do not have access to $\mathbb{P}^\star$ but to $n$ independently and identically distributed (i.i.d.) observations $\mathcal{D} := \xi_{1:n} \sim \mathbb{P}^\star$. Without knowledge of the DGP, model-based inference considers a family of models $\mathcal{P}_\Theta := \{\mathbb{P}_\theta : \theta \in \Theta\} \subset \mathcal{P}(\Xi)$ where each $\mathbb{P}_\theta$ has probability density function $p(\xi|\theta)$ for parameter space $\Theta \subseteq \mathbb{R}^k$. In a Bayesian framework, data $\mathcal{D}$ is combined with a prior $\pi(\theta)$ to obtain posterior beliefs about $\theta$ through $\Pi(\theta|\mathcal{D})$. Bayesian Risk Optimisation (Wu et al., 2018) then solves a stochastic optimisation problem:

$$\min_{x \in \mathbb{R}^d} \mathbb{E}_{\theta \sim \Pi(\theta|\mathcal{D})} \left[ \mathbb{E}_{\xi \sim \mathbb{P}_\theta}[f(x, \xi)] \right]. \tag{1}$$

However, our Bayesian estimator is likely different from the true DGP due to model and data uncertainty: the number of observations may be small; the data noisy; or the prior or model may be misspecified. The optimisation problem (1) inherits any estimation error, and leads to overly optimistic decisions on out-of-sample scenarios even if the estimator is unbiased: this phenomenon is

---

[*]These authors contributed equally to this work.

Workshop on Bayesian Decision-making and Uncertainty, 38th Conference on Neural Information Processing Systems (NeurIPS 2024).

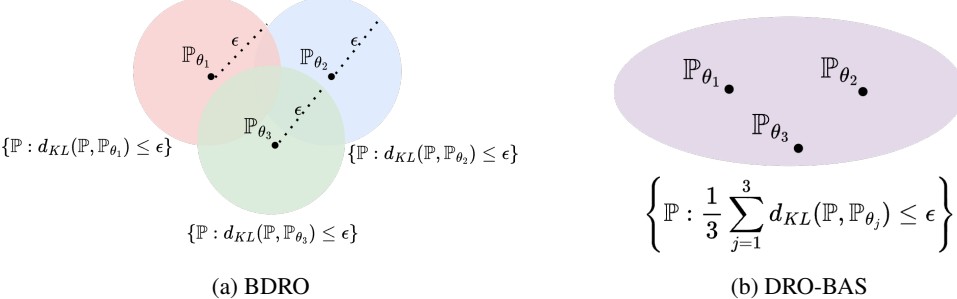

(a) BDRO  (b) DRO-BAS

Figure 1: Illustration of the construction of the BDRO and DRO-BAS optimisation problems for three i.i.d. posterior samples $\theta_1, \theta_2, \theta_3 \sim \Pi(\theta \mid \mathcal{D})$. BDRO seeks the decision that minimises the average worst-case risk between the three ambiguity sets shown in figure (a) whereas DRO-BAS targets the decision minimising the worst-case risk over the ambiguity set shown in (b).

called the optimiser's curse (Kuhn et al., 2019). For example, if the number of observations is small and the prior is overly concentrated, then the decision is likely to be overly optimistic.

To hedge against the uncertainty of the estimated distribution, the field of Distributionally Robust Optimisation (DRO) minimises the expected objective function under the worst-case distribution that lies in an ambiguity set $U \subset \mathcal{P}(\Xi)$. Discrepancy-based ambiguity sets contain distributions that are close to a nominal distribution in the sense of some discrepancy measure such as the Kullback-Leibler (KL) divergence (Hu and Hong, 2013), Wasserstein distance (Kuhn et al., 2019) or Maximum Mean Discrepancy (Staib and Jegelka, 2019). For example, some model-based methods (Iyengar et al., 2023; Michel et al., 2021, 2022) consider a family of parametric models and create discrepancy-based ambiguity sets centered on the fitted model. However, uncertainty about the parameters is not captured in these works, which can lead to a nominal distribution far away from the DGP when the data is limited. The established framework for capturing such uncertainty is Bayesian inference.

The closest work to ours, using parametric Bayesian inference to inform the optimisation problem, is Bayesian DRO (BDRO) by Shapiro et al. (2023). BDRO constructs discrepancy-based ambiguity sets with the KL divergence and takes an *expected worst-case* approach, under the posterior distribution. More specifically, let $U_\epsilon(\mathbb{P}_\theta) := \{\mathbb{Q} \in \mathcal{P}(\Xi) : d_{\mathrm{KL}}(\mathbb{Q}\|\mathbb{P}_\theta) \leq \epsilon\}$ be the ambiguity set centered on distribution $\mathbb{P}_\theta$ with parameter $\epsilon \in [0, \infty)$ controlling the size of the ambiguity set. Under the expected value of the posterior, Bayesian DRO solves:

$$\min_{x \in \mathbb{R}^d} \mathbb{E}_{\theta \sim \Pi(\theta \mid \mathcal{D})} \left[ \sup_{\mathbb{Q} \in U_\epsilon(\mathbb{P}_\theta)} \mathbb{E}_{\xi \sim \mathbb{Q}}[f(x, \xi)] \right], \qquad (2)$$

where $\mathbb{E}_{\theta \sim \Pi(\theta \mid \mathcal{D})}[Y] := \int_\Theta Y(\theta)\Pi(\theta \mid \mathcal{D})\,d\theta$ denotes the expectation of random variable $Y : \Theta \to \mathbb{R}$ with respect to $\Pi(\theta \mid \mathcal{D})$. A decision maker is often interested in protecting against and quantifying the worst-case risk, but BDRO does not correspond to a worst-case risk analysis. Moreover, the BDRO dual problem is a two-stage stochastic problem that involves a double expectation over the posterior and likelihood. To get a good approximation of the dual problem, a large number of samples are required, which increases the solve time of the dual problem.

We introduce DRO with Bayesian Ambiguity Sets (DRO-BAS), an alternative optimisation objective for Bayesian decision-making under uncertainty, based on a posterior-informed ambiguity set. The resulting problem corresponds to a worst-case risk minimisation over distributions with small expected deviation from the candidate model. We go beyond ball-based ambiguity sets, which are dependent on a single nominal distribution, by *allowing the shape of the ambiguity set to be informed by the posterior*. For many exponential family models, we show that the dual formulation of DRO-BAS is an efficient single-stage stochastic program.

## 2 DRO with Bayesian Ambiguity Sets

We propose the following DRO-BAS objective:

$$\min_{x \in \mathbb{R}^d} \sup_{\mathbb{Q}: \mathbb{E}_{\theta \sim \Pi}[D(\mathbb{Q}, \mathbb{P}_\theta)] \leq \epsilon} \mathbb{E}_{\xi \sim \mathbb{Q}} [f_x(\xi)], \tag{3}$$

where $\mathbb{Q} \in \mathcal{P}(\Xi)$ is a distribution in the ambiguity set, $f_x(\xi) := f(x, \xi)$ is the objective function, $D : \mathcal{P}(\Xi) \times \mathcal{P}(\Xi) \to \mathbb{R}$ is a divergence, and $\epsilon \in [0, \infty)$ is a tolerance level. The ambiguity set is informed by the posterior distribution $\Pi$ by considering all probability measures $\mathbb{Q} \in \mathcal{P}(\Xi)$ which are $\epsilon$-away from $\mathbb{P}_\theta$ in expectation, with $\epsilon$ dictating the desired amount of risk in the decision.

The shape of our ambiguity set is flexible and driven by the posterior distribution. This is contrary to standard ambiguity sets which correspond to a ball around a fixed nominal distribution. The DRO-BAS problem (3) is still a worst-case approach, keeping with DRO tradition, instead of BDRO's expected worst-case formulation (2), see Figure 1.

The Bayesian posterior $\Pi(\theta \mid \mathcal{D})$ targets the KL minimiser between the model family and $\mathbb{P}^\star$ (Walker, 2013), hence it is natural to choose $D(\mathbb{Q}, \mathbb{P}_\theta)$ to be the KL divergence of $\mathbb{Q}$ with respect to $\mathbb{P}_\theta$ denoted by $d_{\mathrm{KL}}(\mathbb{Q} \| \mathbb{P}_\theta)$. This means that as $n \to \infty$ the posterior collapses to $\theta_0 := \arg \min_{\theta \in \Theta} d_{\mathrm{KL}}(\mathbb{P}^\star, \mathbb{P}_\theta)$ and the ambiguity set is just a KL-ball around $\mathbb{P}_{\theta_0}$. Using the KL divergence in the DRO-BAS problem in (3), it is straight-forward to obtain an upper bound of the worst-case risk for general models (see Appendix B.1 for a proof):

$$\sup_{\mathbb{Q}: \mathbb{E}_{\theta \sim \Pi}[d_{\mathrm{KL}}(Q \| \mathbb{P}_\theta)] \leq \epsilon} \mathbb{E}_{\xi \sim \mathbb{Q}}[f_x(\xi)] \leq \inf_{\gamma \geq 0} \gamma \epsilon + \mathbb{E}_{\theta \sim \Pi} \left[ \gamma \ln \mathbb{E}_{\xi \sim \mathbb{P}_\theta} \left[ \exp \left( \frac{f_x(\xi)}{\gamma} \right) \right] \right]. \tag{4}$$

Exact closed-form solutions of DRO-BAS can be obtained for a wide range of exponential family models with conjugate priors. When the likelihood distribution is a member of the exponential family, a conjugate prior also belongs to the exponential family (Gelman et al., 1995, Ch. 4.2). In this setting, before we prove the main result, we start with an important Lemma.

**Lemma 1.** *Let $p(\xi \mid \theta)$ be an exponential family likelihood and $\pi(\theta)$, $\Pi(\theta \mid \mathcal{D})$ a conjugate prior-posterior pair, also members of the exponential family. Let $\tau_0, \tau_n \in T$ be hyperparameters of the prior and posterior respectively, where $T$ is the hyperparameter space. Let $\bar{\theta}_n \in \Theta$ depend upon $\tau_n$ and let $G : T \to \mathbb{R}$ be a function of the hyperparameters. If the following identity holds:*

$$\mathbb{E}_{\theta \sim \Pi} [\ln p(\xi \mid \theta)] = \ln p(\xi \mid \bar{\theta}_n) - G(\tau_n), \tag{5}$$

*then the expected KL-divergence can be written as:*

$$\mathbb{E}_{\theta \sim \Pi} [d_{KL}(\mathbb{Q} \| \mathbb{P}_\theta)] = d_{KL}(\mathbb{Q}, \mathbb{P}_{\bar{\theta}_n}) + G(\tau_n). \tag{6}$$

The condition in (5) is a natural property of many exponential family models, some of which are showcased in Table 1. Future work aims to prove this for all exponential family models. It is straight-forward to establish the minimum tolerance level $\epsilon_{\min}$ required to obtain a non-empty ambiguity set. Since the KL divergence is non-negative, under the condition of Lemma 1, for any $\mathbb{Q} \in \mathcal{P}(\Xi)$:

$$\mathbb{E}_{\theta \sim \Pi}[d_{\mathrm{KL}}(\mathbb{Q} \| \mathbb{P}_\theta)] = d_{\mathrm{KL}}(\mathbb{Q}, \mathbb{P}_{\bar{\theta}_n}) + G(\tau_n) \geq G(\tau_n) := \epsilon_{\min}(n). \tag{7}$$

We are now ready to prove our main result.

**Theorem 1.** *Suppose the conditions of Lemma 1 hold and $\epsilon \geq \epsilon_{\min}(n)$ as in (7). Let $\tau_n \in T$, $\bar{\theta}_n \in \Theta$, and $G : T \to \mathbb{R}$. Then*

$$\sup_{\mathbb{Q}: \mathbb{E}_{\theta \sim \Pi}[d_{KL}(\mathbb{Q} \| \mathbb{P}_\theta)] \leq \epsilon} \mathbb{E}_{\xi \sim \mathbb{Q}}[f_x(\xi)] = \inf_{\gamma \geq 0} \gamma(\epsilon - G(\tau_n)) + \gamma \ln \mathbb{E}_{\xi \sim p(\xi \mid \bar{\theta}_n)} \left[ \exp \left( \frac{f_x(\xi)}{\gamma} \right) \right]. \tag{8}$$

To guarantee that the DRO-BAS objective upper bounds the expected risk under the DGP, the decision-maker aims to choose $\epsilon$ large enough so that $\mathbb{P}^\star$ is contained in the ambiguity set. The condition in (5) yields a closed-form expression for the optimal radius $\epsilon^\star$ by noting that:

$$\epsilon^\star = \mathbb{E}_{\theta \sim \Pi}[d_{\mathrm{KL}}(\mathbb{P}^\star \| \mathbb{P}_\theta)] = d_{\mathrm{KL}}(\mathbb{P}^\star, \mathbb{P}_{\bar{\theta}_n}) + G(\tau_n). \tag{9}$$

If the model is well-specified, and hence $\mathbb{P}^\star$ and $\mathbb{P}_{\bar{\theta}_n}$ belong to the same exponential family, it is straightforward to obtain $\epsilon^\star$ based on the prior, posterior and true parameter values. We give examples in Appendix A. In practice, since the true parameter values are unknown, we can approximate $\epsilon^\star$ using the observed samples. It follows that for any $\epsilon \geq \epsilon^\star \geq \epsilon_{\min}(n)$:

$$\mathbb{E}_{\xi \sim \mathbb{P}^\star}[f(x, \xi)] \leq \sup_{\mathbb{Q}: \mathbb{E}_{\theta \sim \Pi}[d_{\mathrm{KL}}(Q \| \mathbb{P}_\theta)] \leq \epsilon} \mathbb{E}_{\xi \sim \mathbb{Q}}[f_x(\xi)].$$

Table 1: Examples for Theorem 1 of the parameter $\bar{\theta}_n$ and the function $G(\tau_n)$ for different likelihoods $p(\xi \mid \theta)$ with conjugate posterior $\Pi(\theta \mid \tau_n)$ and posterior hyperparameters $\tau_n$. The normal, normal-gamma, exponential, and gamma distributions are denoted $\mathcal{N}$, NG, Exp, and Ga respectively. See the supplementary material for the definitions of $\tau_n$ and the derivations of $\bar{\theta}_n$ and $G(\tau_n)$.

| $p(\xi \mid \theta)$ | $\Pi(\theta \mid \tau_n)$ | $\bar{\theta}_n$ | $G(\tau_n)$ |
|---|---|---|---|
| $\mathcal{N}(\xi \mid \mu, \sigma^2)$ | $\mathcal{N}(\mu \mid \mu_n, \sigma_n^2)$ | $\mu_n, \sigma^2$ | $\frac{\sigma_n^2}{2\sigma^2}$ |
| $\mathcal{N}(\xi \mid \mu, \lambda^{-1})$ | $\mathrm{NG}(\mu, \lambda \mid \mu_n, \kappa_n, \alpha_n, \beta_n)$ | $\mu_n, \frac{\beta_n}{\alpha_n}$ | $\frac{1}{2}\left(\frac{1}{\kappa_n} + \ln\alpha_n - \psi(\alpha_n)\right)$ |
| $\mathrm{Exp}(\xi \mid \theta)$ | $\mathrm{Ga}(\theta \mid \alpha_n, \beta_n)$ | $\frac{\alpha_n}{\beta_n}$ | $\ln\alpha_n - \psi(\alpha_n)$ |

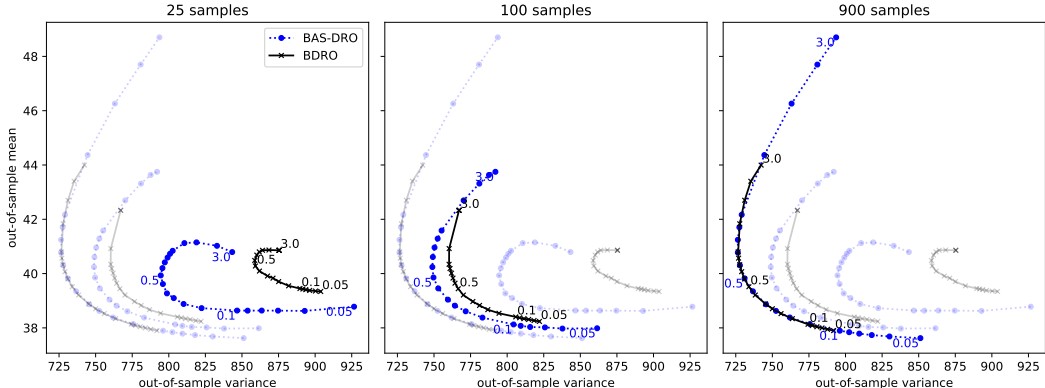

Figure 2: The out-of-sample mean-variance tradeoff (in bold) while varying $\epsilon$ for DRO-BAS and BDRO when the total number of samples from the model is 25 (left), 100 (middle), and 900 (right).

## 3 The Newsvendor Problem

**Experiment setup.** We evaluate DRO-BAS against the BDRO framework on a univariate Newsvendor problem with a well-specified univariate Gaussian likelihood with unknown mean and variance (Appendix D showcases a misspecified setting). The goal is to choose an inventory level $0 \le x \le 50$ of a perishable product with unknown customer demand $\xi \in \mathbb{R}$ that minimises the cost function $f(x, \xi) = h \max(0, x - \xi) + b \max(0, \xi - x)$, where $h$ and $b$ are the holding cost and backorder cost per unit of the product respectively. We let $\mathbb{P}^\star$ be a univariate Gaussian $\mathcal{N}(\mu_\star, \sigma_\star^2)$ with $\mu_\star = 25$ and $\sigma_\star^2 = 100$. For random seed $j = 1, \dots, 200$, the training dataset $\mathcal{D}_n^{(j)}$ contains $n = 20$ observations and the test dataset $\mathcal{D}_m^{(j)}$ contains $m = 50$ observations. The conjugate prior and posterior are normal-gamma distributions (Appendix A.2). $N$ is the total number of samples from each model. For each seed $j$, we run DRO-BAS and BDRO with $N = 25, 100, 900$ and across 21 different values of $\epsilon$ ranging from 0.05 to 3. For DRO-BAS, $N$ is the number of samples from $p(\xi \mid \bar{\theta}_n)$ and for BDRO, $N = N_\theta \times N_\xi$ where $N_\theta$ is the number of posterior samples and $N_\xi$ likelihood samples due to the double expectation present; we set $N_\theta = N_\xi$ to compare models on an equal $N$ total samples regime. For a given $\epsilon$, we calculate the out-of-sample mean $m(\epsilon)$ and variance $v(\epsilon)$ of the objective function $f(x_\epsilon^{(j)}, \hat{\xi}_i)$ over all $\hat{\xi}_i \in \mathcal{D}_m^{(j)}$ and over all seeds $j = 1, \dots, 200$, where $x_\epsilon^{(j)}$ is the optimal solution on training dataset $\mathcal{D}_n^{(j)}$ (see Appendix C).

**Analysis.** Figure 2 shows that, for small sample size $N = 25, 100$, our framework *dominates* BDRO in the sense that DRO-BAS forms a Pareto front for the out-of-sample mean-variance tradeoff of the objective function $f$. That is, for any $\epsilon_1$, let $m_{\mathrm{BDRO}}(\epsilon_1)$ and $v_{\mathrm{BDRO}}(\epsilon_1)$ be the out-of-sample mean and variance respectively of BDRO: then there exists $\epsilon_2$ with out-of-sample mean $m_{\mathrm{BAS}}(\epsilon_2)$ and variance $v_{\mathrm{BAS}}(\epsilon_2)$ of BAS-DRO such that $m_{\mathrm{BAS}}(\epsilon_2) < m_{\mathrm{BDRO}}(\epsilon_1)$ and $v_{\mathrm{BAS}}(\epsilon_2) < v_{\mathrm{BDRO}}(\epsilon_1)$. When $N = 900$, Figure 2 shows DRO-BAS and BDRO lie roughly on the same Pareto front. To summarise, BDRO requires more samples $N$ than DRO-BAS for good out-of-sample performance,

likely because BDRO must evaluate a double expectation over the posterior and likelihood, whilst DRO-BAS only samples from $p(\xi \mid \bar{\theta}_n)$. For fixed $N$, the solve times for DRO-BAS and BDRO are broadly comparable (see Appendix C).

## 4 Discussion

We proposed a novel approach to Bayesian decision-making under uncertainty through a DRO objective based on posterior-informed Bayesian ambiguity sets. The resulting optimisation problem is a single-stage stochastic program with closed-form formulation for a variety of exponential-family models. The suggested methodology has good out-of-sample performance, as showcased in Figure 2. In our recent work (Dellaporta et al., 2024), we have also extended DRO-BAS to a general formulation for exponential family models and investigated alternative Bayesian ambiguity sets based on the posterior predictive distribution. Finally, whilst DRO-BAS offers protection against distributional ambiguity, the dependence of DRO-BAS on the Bayesian posterior makes it vulnerable to model misspecification. Future work will explore robust Bayesian ambiguity sets that address model misspecification through robust posteriors and discrepancies.

## Acknowledgments and Disclosure of Funding

CD acknowledges support from EPSRC grant [EP/T51794X/1] as part of the Warwick CDT in Mathematics and Statistics. PO and TD acknowledge support from a UKRI Turing AI acceleration Fellowship [EP/V02678X/1] and a Turing Impact Award from the Alan Turing Institute. For the purpose of open access, the authors have applied a Creative Commons Attribution (CC-BY) license to any Author Accepted Manuscript version arising from this submission.

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

# Supplementary Material

The Supplementary Material is organised as follows: Appendix A provides details for all the exponential family models discussed in Table 1, while Appendix B contains the proofs of all mathematical results appearing in the main text. Appendix C provides additional experimental details for the Newsvendor problem in Section 3. Finally, in Appendix D we present experimental results for the newsvendor problem example with a misspecified model.

## A Special cases

We derive the values of $G(\tau_n)$ and $\bar{\theta}_n$ for different likelihoods and conjugate prior/posterior in Table 1. Each subsection contains a corollary with the result in Table 1.

### A.1 Gaussian model with unknown mean and known variance

Let the random variable $\xi$ be univariate and have continuous support. We assume the variance $\sigma^2$ of $\xi$ is known. We estimate the mean of a univariate Gaussian distribution with known variance $\sigma^2$. The example can be found in (Bishop, 2006, Section 2.3.6). We define our parameter $\theta$ to be the unknown mean $\mu$ and we place a Gaussian prior $\pi(\mu)$ over it.

**Definition 1** (Gaussian with unknown mean and known variance)**.** *The likelihood is* $p(\xi \mid \mu) = \mathcal{N}(\mu, \sigma^2)$*, the prior over* $\mu$ *is* $\pi(\mu) = \mathcal{N}(\mu_0, \sigma_0^2)$*, and the conjugate posterior is* $\Pi(\mu \mid \mathcal{D}) = \mathcal{N}(\mu \mid \mu_n, \sigma_n^2)$*, where*

$$\mu_n := \frac{\sigma^2}{n\sigma_0^2 + \sigma^2}\mu_0 + \frac{n\sigma_0^2}{n\sigma_0^2 + \sigma^2}\hat{\mu} \qquad \frac{1}{\sigma_n^2} := \frac{1}{\sigma_0^2} + \frac{n}{\sigma^2} \qquad \hat{\mu} := \frac{1}{n}\sum_{i=1}^{n}\xi_i.$$

**Lemma 2.** *Let* $\mu_n \in \mathbb{R}$ *and* $\sigma, \sigma_n \in \mathbb{R}_+$*. Then*

$$\mathbb{E}_{\mu \sim \mathcal{N}(\mu_n, \sigma_n^2)}\left[\log \mathcal{N}(\mu, \sigma^2)\right] = \log \mathcal{N}(\mu_n, \sigma^2) - \frac{\sigma_n^2}{2\sigma^2}.$$

*Proof.* The result is a special case of (Stratos, 2023, Lemma H.10). $\qquad\square$

**Corollary 1.** *When the likelihood is a Gaussian distribution with unknown mean and known variance* $\sigma^2$ *and the prior and posterior are Gaussian distributions (see Definition 1), then Theorem 1 holds with* $\bar{\theta}_n = \mu_n$ *and* $G(\tau_n) = \frac{\sigma_n^2}{2\sigma^2}$*.*

*Proof.* Lemma 2 shows that the condition (5) in Lemma 1 holds, thus Theorem 1 follows. $\qquad\square$

**Tolerance level** $\epsilon$  In the well-specified case, where we assume that $\mathbb{P}^\star := \mathbb{P}_\theta^\star$ for some $\theta^\star \in \Theta$, it is easy to obtain the required size of the ambiguity set exactly. Let $\theta^\star := \mu^\star$ and $\mathbb{P}^\star := N(\mu^\star, \sigma^2)$. By Corollary 1 it follows that:

$$\mathbb{E}_{\theta \sim N(\mu_n, \sigma_n^2)}\left[d_{\mathrm{KL}}(\mathbb{P}^\star, \mathbb{P}_\theta)\right] = \frac{(\mu^\star - \mu_n)^2 + \sigma_n^2}{2\sigma^2}. \tag{10}$$

So for a fixed finite sample $\xi_{1:n} \sim \mathbb{P}^\star$, if $\epsilon \geq \epsilon^\star := \frac{(\mu^\star - \mu_n)^2 + \sigma_n^2}{2\sigma^2}$, it follows that DRO-BAS upper-bounds the target optimisation objective:

$$\mathbb{E}_{\xi \sim \mathbb{P}^\star}[f_x(\xi)] \leq \sup_{\mathbb{Q}:\mathbb{E}_{\theta \sim \Pi(\cdot|\xi_{1:n})}[d_{\mathrm{KL}}(Q \parallel \mathbb{P}_\theta)] \leq \epsilon} \mathbb{E}_{\xi \sim \mathbb{Q}}[f_x(\xi)].$$

In practice, since $\mu^\star$ is unknown, $\epsilon^\star$ can be approximated by using the sample mean.

## A.2 Gaussian Model with Unknown Mean and Variance

In this section, we consider a Bayesian model that estimates the unknown mean and variance of a uni-variate Gaussian distribution. We define our model in Definition 2, then prove some preliminary results for the normal-gamma distribution, before proving our main result in Lemma 4.

**Definition 2** (Unknown mean and variance of a Gaussian). *Following (Murphy, 2023, Chapter 3.4.3), we place a normal-gamma prior over the mean $\mu$ and precision $\lambda = \sigma^{-2}$. The normal-gamma prior is the conjugate to a Gaussian likelihood and results in a normal-gamma posterior distribution.*

*Likelihood:* $\quad p(\xi \mid \mu, \lambda) = \mathcal{N}(\xi \mid \mu, \lambda^{-1})$

*Prior:* $\qquad \pi(\mu, \lambda) = NG(\mu, \lambda \mid \mu_0, \kappa_0, \alpha_0, \beta_0) = \mathcal{N}(\mu \mid \mu_0, (\kappa_0 \lambda)^{-1}) \cdot Ga(\lambda \mid \alpha_0, \beta_0)$

*Posterior:* $\quad \pi(\mu, \lambda \mid \mathcal{D}) = NG(\mu, \lambda \mid \mu_n, \kappa_n, \alpha_n, \beta_n) = \mathcal{N}(\mu \mid \mu_n, (\lambda \kappa_n)^{-1}) \cdot Ga(\lambda \mid \alpha_n, \beta_n)$

*where*

$$\mu_n := \frac{\kappa_0 \mu_0 + n\bar{\xi}_n}{n + \kappa_0}, \qquad \kappa_n := \kappa_0 + n, \qquad \alpha_n := \alpha_0 + \frac{n}{2}, \qquad \bar{\xi}_n = \frac{1}{n}\sum_{i-1}^{n}\xi_i,$$

$$\beta_n := \beta_0 + \frac{1}{2}\sum_{i=1}^{n}(\xi_i - \bar{\xi}_n)^2 + \frac{\kappa_0 n (\bar{\xi}_n - \mu_0)^2}{2(\kappa_0 + n)}.$$

In Lemma 4, we derive condition Equation (5) for a Gaussian model with unknown mean and unknown variance. Before proceeding, we need to define the gamma and digamma functions and recall the moments of the normal-gamma distribution.

**Definition 3.** *The gamma function $\Gamma : \mathbb{N} \to \mathbb{R}$ and digamma function $\psi : \mathbb{N} \to \mathbb{R}$ are*

$$\Gamma(z) := (z-1)! \qquad\qquad \psi(z) := \frac{\mathrm{d}}{\mathrm{d}z}\ln\Gamma(z).$$

**Lemma 3.** *Let $NG(\mu, \lambda \mid \mu_n, \kappa_n, \alpha_n, \beta_n)$ be a normal-gamma distribution with parameters $\mu_n \in \mathbb{R}$ and $\kappa_n, \alpha_n, \beta_n \in \mathbb{R}_+$. The moments of the normal-gamma distribution are*

$$\mathbb{E}_{NG}[\ln\lambda] = \psi(\alpha_n) - \ln\beta_n, \quad \mathbb{E}_{NG}[\lambda] = \frac{\alpha_n}{\beta_n}, \quad \mathbb{E}_{NG}[\lambda\mu] = \mu_n\frac{\alpha_n}{\beta_n}, \quad \mathbb{E}_{NG}[\lambda\mu^2] = \frac{1}{\kappa_n} + \mu_n^2\frac{\alpha_n}{\beta_n}.$$

*where $\psi : \mathbb{N} \to \mathbb{R}$ is the digamma function from Definition 3.*

**Lemma 4.** *Let $\mu_n \in \mathbb{R}$ and $\kappa_n, \alpha_n, \beta_n \in \mathbb{R}_+$. Then*

$$\mathbb{E}_{NG(\mu,\lambda\mid\mu_n,\kappa_n,\alpha_n,\beta_n)}\left[\ln\mathcal{N}(\xi \mid \mu, \lambda^{-1})\right] = \ln\mathcal{N}\left(\xi \mid \mu_n, \frac{\beta_n}{\alpha_n}\right) - \frac{1}{2}\left(\frac{1}{\kappa_n} + \ln\alpha_n - \psi(\alpha_n)\right)$$

*where $\psi : \mathbb{N} \to \mathbb{R}$ is the digamma function from Definition 3.*

*Proof.* First, observe that the natural logarithm of the Gaussian distribution may be re-written as

$$\ln\mathcal{N}(\xi \mid \mu, \lambda^{-1}) = \ln\left(\frac{\lambda^{\frac{1}{2}}}{(2\pi)^{\frac{1}{2}}}\exp\left(-\frac{\lambda}{2}(\xi - \mu)^2\right)\right)$$

$$= \frac{1}{2}\ln\lambda - \frac{1}{2}\ln 2\pi - \frac{\lambda}{2}(\xi - \mu)^2$$

$$= \frac{1}{2}\ln\lambda - \frac{1}{2}\ln 2\pi - \frac{1}{2}\lambda\xi^2 + \lambda\mu\xi - \frac{1}{2}\lambda\mu^2.$$

In what follows, for shorthand, we denote the expectation $\mathbb{E}_{NG(\mu,\lambda\mid\mu_n,\kappa_n,\alpha_n,\beta_n)}$ as $\mathbb{E}_{\mu,\lambda\sim NG}$:

$$\mathbb{E}_{\mu,\lambda\sim NG}\left[\ln\mathcal{N}(\xi\mid\mu,\lambda^{-1})\right]$$

$$\overset{(i)}{=}\mathbb{E}_{\mu,\lambda\sim NG}\left[\frac{1}{2}\ln\lambda-\frac{1}{2}\ln 2\pi-\frac{1}{2}\lambda\xi^2+\lambda\mu\xi-\frac{1}{2}\lambda\mu^2\right]$$

$$\overset{(ii)}{=}-\frac{1}{2}\ln 2\pi+\frac{1}{2}\mathbb{E}_{\mu,\lambda\sim NG}\left[\ln\lambda\right]-\frac{1}{2}\xi^2\cdot\mathbb{E}_{\mu,\lambda\sim NG}\left[\lambda\right]+\xi\cdot\mathbb{E}_{\mu,\lambda\sim NG}\left[\lambda\mu\right]-\frac{1}{2}\mathbb{E}_{\mu,\lambda\sim NG}\left[\lambda\mu^2\right]$$

$$\overset{(iii)}{=}-\frac{1}{2}\ln 2\pi+\frac{1}{2}(\psi(\alpha_n)-\ln\beta_n)-\frac{1}{2}\xi^2\frac{\alpha_n}{\beta_n}+\xi\mu_n\frac{\alpha_n}{\beta_n}-\frac{1}{2}\left(\frac{1}{\kappa_n}+\mu_n^2\frac{\alpha_n}{\beta_n}\right)$$

$$\overset{(iv)}{=}-\frac{1}{2}\ln 2\pi+\frac{1}{2}\left(\psi(\alpha_n)-\ln\beta_n-\frac{1}{\kappa_n}\right)-\frac{\alpha_n}{2\beta_n}\left(\xi-\mu_n\right)^2$$

$$\overset{(v)}{=}-\frac{1}{2}\ln 2\pi-\frac{1}{2}\left(\ln\beta_n-\ln\alpha_n+\ln\alpha_n-\psi(\alpha_n)+\frac{1}{\kappa_n}\right)+\ln\exp\left(-\frac{1}{2\frac{\beta_n}{\alpha_n}}\left(\xi-\mu_n\right)^2\right)$$

$$\overset{(vi)}{=}-\frac{1}{2}\left(\ln\alpha_n-\psi(\alpha_n)+\frac{1}{\kappa_n}\right)-\frac{1}{2}\ln\left(2\pi\frac{\beta_n}{\alpha_n}\right)+\ln\exp\left(-\frac{1}{2\frac{\beta_n}{\alpha_n}}\left(\xi-\mu_n\right)^2\right)$$

$$\overset{(vii)}{=}-\frac{1}{2}\left(\frac{1}{2\alpha_n}+I_{\alpha_n}+\frac{1}{\kappa_n}\right)+\ln\left(\frac{1}{\sqrt{2\pi\frac{\beta_n}{\alpha_n}}}\exp\left(-\frac{1}{2\frac{\beta_n}{\alpha_n}}(\xi-\mu_n)^2\right)\right)$$

$$\overset{(viii)}{=}-\frac{1}{2}\left(\ln\alpha_n-\psi(\alpha_n)+\frac{1}{\kappa_n}\right)+\ln\mathcal{N}\left(\xi\mid\mu_n,\frac{\beta_n}{\alpha_n}\right)$$

where in equation (i) we take the expectation over the normal-gamma distribution; (ii) we apply linearity of expectation; (iii) we use the moment-generating functions from Lemma 3; (iv) we complete the square; (v) we add and subtract $\ln\alpha_n$; (vi) and (vii) we re-arrange and apply log identities; and finally in (viii) we use the definition of a Gaussian probability density function. □

**Corollary 2.** *When the likelihood is a Gaussian distribution with unknown mean and variance, and the conjugate prior and posterior are normal-gamma distributions (see Definition 2), then Theorem 1 holds with $\bar{\theta}_n=(\mu_n,\frac{\beta_n}{\alpha_n})$ and $G(\tau_n)=\frac{1}{2}\left(\ln\alpha_n-\psi(\alpha_n)+\frac{1}{\kappa_n}\right)$.*

*Proof.* Lemma 4 shows that the condition (5) in Lemma 1 holds, thus Theorem 1 follows. □

**Tolerance level $\epsilon$** In the well-specified case, where we assume that $\mathbb{P}^\star:=\mathbb{P}_\theta^\star$ for some $\theta^\star\in\Theta$, it is easy to obtain the required size of the ambiguity set exactly. Let $\theta^\star:=(\mu^\star,\lambda^{\star-1})$ and $\mathbb{P}^\star:=N(\mu^\star,\lambda^{\star-1})$. Using Corollary 2 we obtain:

$$\mathbb{E}_{\mu,\lambda\sim NG(\mu,\lambda\mid\mu_n,\kappa_n,\alpha_n,\beta_n)}\left[d_{\mathrm{KL}}(\mathbb{P}^\star,\mathcal{N}(\xi\mid\mu,\lambda^{-1}))\right]$$

$$=d_{\mathrm{KL}}\left(\mathbb{P}^\star\parallel\mathcal{N}\left(\mu_n,\frac{\beta_n}{\alpha_n}\right)\right)+\frac{1}{2}\left(\frac{1}{\kappa_n}+\ln\alpha_n-\psi(\alpha_n)\right)$$

$$=\ln\left(\sqrt{\lambda^\star\frac{\beta_n}{\alpha_n}}\right)+\frac{\lambda^{\star-1}+(\mu^\star-\mu_n)^2}{2\frac{\beta_n}{\alpha_n}}-\frac{1}{2}+\frac{1}{2}\left(\frac{1}{\kappa_n}+\ln\alpha_n-\psi(\alpha_n)\right)$$

$$=\frac{1}{2}\left(\ln\left(\lambda^\star\beta_n\right)+\frac{\lambda^{\star-1}+(\mu^\star-\mu_n)^2)}{2\frac{\beta_n}{\alpha_n}}-1+\frac{1}{\kappa_n}-\psi(\alpha_n)\right).$$

### A.3  Exponential likelihood with conjugate gamma prior

**Definition 4.** *The likelihood $p(\xi\mid\theta)$ is an exponential distribution $\mathrm{Exp}(\xi\mid\theta)$ where $\theta>0$ is the rate parameter. The prior $\pi(\theta)$ is a gamma distribution $\mathrm{Ga}(\theta\mid\alpha_0,\beta_0)$ with shape $\alpha_0>0$ and rate $\beta_0>0$. The parameters $\alpha_n,\beta_n$ of the posterior $\pi(\theta\mid\mathcal{D})=\mathrm{Ga}(\theta\mid\alpha_n,\beta_n)$ are given by $\alpha_n=\alpha_0+n$ and $\beta_n=\beta_0+\sum_{\xi_i\in\mathcal{D}}\xi_i$.*

**Lemma 5.** *When the likelihood is an exponential distribution with gamma prior and posterior (see Definition 4), then*

$$\mathbb{E}_{\mathrm{Ga}(\theta|\alpha_n,\beta_n)}\left[\ln\mathrm{Exp}(\xi\mid\theta)\right]=\ln\mathrm{Exp}\left(\xi\mid\frac{\alpha_n}{\beta_n}\right)+\psi(\alpha_n)-\ln\alpha_n.$$

*Proof.* Starting from the left-hand side, we take the log of the PDF of the exponential distribution, then use the logarithm expectation of the gamma distribution, and finally re-arrange using log identities:

$$
\begin{aligned}
\mathbb{E}_{\mathrm{Ga}(\theta|\alpha_n,\beta_n)}\left[\ln\mathrm{Exp}(\xi\mid\theta)\right] &= \mathbb{E}_{\mathrm{Ga}(\theta|\alpha_n,\beta_n)}\left[\ln\theta-\theta\xi\right]\\
&= \psi(\alpha_n)-\ln\beta_n-\frac{\alpha_n}{\beta_n}\xi\\
&= \psi(\alpha_n)-\ln\alpha_n+\ln\frac{\alpha_n}{\beta_n}-\frac{\alpha_n}{\beta_n}\xi\\
&= \psi(\alpha_n)-\ln\alpha_n+\ln\left(\frac{\alpha_n}{\beta_n}\exp\left(-\frac{\alpha_n}{\beta_n}\xi\right)\right)\\
&= \psi(\alpha_n)-\ln\alpha_n+\ln\mathrm{Exp}\left(\xi\mid\frac{\alpha_n}{\beta_n}\right).
\end{aligned}
$$

The last line follows by the definition of the PDF of the exponential distribution. $\square$

**Corollary 3.** *When the likelihood is an exponential distribution with gamma prior and posterior, then Theorem 1 holds with $\bar{\theta}_n=\frac{\alpha_n}{\beta_n}$ and $G(\tau_n)=\psi(\alpha_n)-\ln\alpha_n$.*

*Proof.* Lemma 5 shows that the condition (5) in Lemma 1 holds, thus Theorem 1 follows. $\square$

**Tolerance level $\epsilon$** In the well-specified case, where we assume that $\mathbb{P}^\star:=\mathbb{P}_\theta^\star$ for some $\theta^\star\in\Theta$, it is easy to obtain the required size of the ambiguity set exactly. Let $\theta^\star$ be the true rate parameter, i.e. $\mathbb{P}^\star:=\mathrm{Exp}(\theta^\star)$. Using Corollary 3 we obtain:

$$
\begin{aligned}
&\mathbb{E}_{\mathrm{Ga}(\theta|\alpha_n,\beta_n)}\left[d_{\mathrm{KL}}(\mathbb{P}^\star,\mathrm{Exp}(\theta)\right]\\
&= d_{\mathrm{KL}}\left(\mathbb{P}^\star\parallel\mathrm{Exp}\left(\frac{\alpha_n}{\beta_n}\right)\right)+\psi(\alpha_n)-\ln(\alpha_n)\\
&= \ln(\theta^\star)-\ln\left(\frac{\alpha_n}{\beta_n}\right)+\frac{\alpha_n}{\beta_n\theta^\star}-1+\psi(\alpha_n)-\ln(\alpha_n).
\end{aligned}
$$

# B  Proofs of Theoretical Results

## B.1  Proofs of DRO-BAS upper bound in Equation (4)

Before proving the required upper bound, we recall the definition of the KL divergence and its convex conjugate.

**Definition 5** (KL-divergence). *Let $\mu,\nu\in\mathcal{P}(\Xi)$ and assume $\mu$ is absolutely continuous with respect to $\nu$ ($\mu\ll\nu$). The $KL$-divergence of $\mu$ with respect to $\nu$ is defined as:*

$$d_{KL}(\mu\|\nu):=\int_\Xi\ln\left(\frac{\mu(d\xi)}{\nu(d\xi)}\right)\mu(d\xi).$$

**Lemma 6** (Conjugate of the KL-divergence). *Let $\nu\in\mathcal{P}(\Xi)$ be non-negative and finite. The convex conjugate $d_{KL}^\star(\cdot\|\nu)$ of $d_{KL}(\cdot\|\nu)$ is*

$$d_{KL}^\star(\cdot\|\nu)(h)=\ln\left(\int_\Xi\exp(h)d\nu\right).$$

*Proof.* See Proposition 28 and Example 7 in Agrawal and Horel (2021). $\square$

*Proof of Equation 4.* The result follows from a standard Lagrangian duality argument and an application of Jensen's inequality. More specifically, we introduce a Lagrangian variable $\gamma \geq 0$ for the expected-ball constraint on the left-hand side of (4) as follows:

$$\sup_{\mathbb{Q}:\mathbb{E}_{\theta\sim\Pi}[d_{\mathrm{KL}}(Q\|\mathbb{P}_\theta)]\leq\epsilon} \mathbb{E}_Q[f_x] \overset{(i)}{\leq} \inf_{\gamma\geq0} \sup_{\mathbb{Q}\in\mathcal{P}(\Xi)} \mathbb{E}_Q[f_x] + \gamma\epsilon - \gamma\mathbb{E}_\Pi\left[d_{\mathrm{KL}}(\mathbb{Q}\|\mathbb{P}_\theta)\right]$$

$$\overset{(ii)}{=} \inf_{\gamma\geq0} \gamma\epsilon + \sup_{\mathbb{Q}\in\mathcal{P}(\Xi)} \mathbb{E}_Q[f_x] - \mathbb{E}_\Pi\left[\gamma d_{\mathrm{KL}}(\mathbb{Q}\|\mathbb{P}_\theta)\right]$$

$$\overset{(iii)}{=} \inf_{\gamma\geq0} \gamma\epsilon + \left(\mathbb{E}_\Pi\left[\gamma d_{\mathrm{KL}}(\cdot\|\mathbb{P}_\theta)\right]\right)^\star (f_x)$$

$$\overset{(iv)}{\leq} \inf_{\gamma\geq0} \gamma\epsilon + \mathbb{E}_\Pi\left[\left(\gamma d_{\mathrm{KL}}(\cdot\|\mathbb{P}_\theta)\right)^\star (f_x)\right]$$

$$\overset{(v)}{=} \inf_{\gamma\geq0} \gamma\epsilon + \mathbb{E}_\Pi\left[\gamma \ln \mathbb{E}_{\mathbb{P}_\theta}\left[\exp\left(\frac{f_x}{\gamma}\right)\right]\right].$$

Inequality (i) holds by weak duality. Equality (ii) holds by linearity of expectation and a simple rearrangement. Equality (iii) holds by the definition of the conjugate function. Inequality (iv) holds by Jensen's inequality $(\mathbb{E}[\cdot])^\star \leq \mathbb{E}[(\cdot)^\star]$ because the conjugate is a convex function. Equality (v) holds by Lemma 6 and the fact that for $\gamma \geq 0$ and function $\phi$, $(\gamma\phi)^\star(y) = \gamma\phi^\star(y/\gamma)$. $\qquad\square$

## B.2 Proof of Lemma 1

Starting from the left-hand side, we have

$$\mathbb{E}_\Pi\left[d_{\mathrm{KL}}(\mathbb{Q}\|\mathbb{P}_\theta)\right] \overset{(i)}{=} \mathbb{E}_{\theta\sim\pi(\theta|\mathcal{D})}\left[\int_\Xi q(\xi)\ln\left(\frac{q(\xi)}{p(\xi\mid\theta)}\right)\mathrm{d}\xi\right]$$

$$\overset{(ii)}{=} \mathbb{E}_{\theta\sim\pi(\theta|\mathcal{D})}\left[\int_\Xi q(\xi)\ln\left(q(\xi)\right) - q(\xi)\ln\left(p(\xi\mid\theta)\right)\mathrm{d}\xi\right]$$

$$\overset{(iii)}{=} \int_\Xi q(\xi)\ln\left(q(\xi)\right) - q(\xi)\cdot\mathbb{E}_{\theta\sim\pi(\theta|\mathcal{D})}\left[\ln\left(p(\xi\mid\theta)\right)\right]\mathrm{d}\xi$$

$$\overset{(iv)}{=} \int_\Xi q(\xi)\ln\left(q(\xi)\right) - q(\xi)\cdot\left(\ln p(\xi\mid\bar\theta_n) - G(\tau_n)\right)\mathrm{d}\xi$$

$$\overset{(v)}{=} \int_\Xi q(\xi)\ln\left(\frac{q(\xi)}{p(\xi\mid\bar\theta_n)}\right)\mathrm{d}\xi + \int_\Xi q(\xi)\cdot G(\tau_n)\mathrm{d}\xi$$

$$\overset{(vi)}{=} d_{\mathrm{KL}}(\mathbb{Q}\|\mathbb{P}_{\bar\theta_n}) + \mathbb{E}_\mathbb{Q}[G(\tau_n)]$$

$$\overset{(vii)}{=} d_{\mathrm{KL}}(q(\xi)\|p(\xi\mid\bar\theta_n)) + G(\tau_n).$$

where (i) is by the definition of the KL-divergence; (ii) follows by $\log$ properties; (iii) holds by linearity of expectation; (iv) holds by condition (5) in Lemma 1; (v) holds by rearrangement and properties of $\log$; (vi) holds by the definition of the KL-divergence and the definition of $\mathbb{E}_\mathbb{Q}$; and (vii) holds by the expected value of a constant.

## B.3 Proof of Theorem 1

We begin by restating the Lagrangian dual from the proof of Equation (4), but with the added claim that strong duality holds between the primal and dual problems:

$$\sup_{\mathbb{Q}:\mathbb{E}_{\theta\sim\Pi}[d_{\mathrm{KL}}(\mathbb{Q}\|\mathbb{P}_\theta)]\leq\epsilon} \mathbb{E}_\mathbb{Q}[f_x] = \inf_{\gamma\geq0} \sup_{\mathbb{Q}\in\mathcal{P}(\Xi)} \mathbb{E}_\mathbb{Q}[f_x] + \gamma\epsilon - \gamma\mathbb{E}_\Pi\left[d_{\mathrm{KL}}(\mathbb{Q}\|\mathbb{P}_\theta)\right]. \tag{11}$$

*Proof.* The conditions under which our claim of strong duality holds will be proved later. Next, we substitute the right-hand side of equation (vi) above into the dual problem in (11):

$$\sup_{\mathbb{Q}:\mathbb{E}_{\theta\sim\Pi}[d_{\mathrm{KL}}(\mathbb{Q}\,\|\,\mathbb{P}_\theta)]\leq\epsilon} \mathbb{E}_{\xi\sim\mathbb{Q}}[f_x(\xi)]$$

$$= \inf_{\gamma\geq 0}\ \gamma\epsilon + \sup_{\mathbb{Q}\in\mathcal{P}(\Xi)}\ \int_\Xi f_x(\xi)q(\xi)\,\mathrm{d}\xi - \gamma\left(d_{\mathrm{KL}}\left(\mathbb{Q}\,\|\,p(\xi\mid\bar\theta_n)\right) + G(\tau_n)\right)$$

$$= \inf_{\gamma\geq 0}\ \gamma\epsilon - \gamma G(\tau_n) + \left(\gamma\,d_{\mathrm{KL}}\left(\cdot\,\|\,p(\xi\mid\bar\theta_n)\right)\right)^\star(f_x(\xi))$$

$$= \inf_{\gamma\geq 0}\ \gamma(\epsilon - G(\tau_n)) + \gamma\ln\mathbb{E}_{\xi\sim p(\xi|\bar\theta_n)}\left[\exp\left(\frac{f_x(\xi)}{\gamma}\right)\right],$$

where the second and third equality holds by the definition of the conjugate of the KL-divergence and by Lemma 6.

Finally, it remains to argue that strong duality holds. First, note that the primal problem is a concave optimisation problem with respect to distribution $\mathbb{Q}$. Second, when $\epsilon > G(\tau_n)$, then distribution $\hat{\mathbb{Q}} = p(\xi\mid\bar\theta_n)$ is a strictly feasible point to the primal constraint because

$$\mathbb{E}_{\theta\sim\Pi}[d_{\mathrm{KL}}(\hat{\mathbb{Q}}\,\|\,\mathbb{P}_\theta)] = 0 < \epsilon - G(\tau_n).$$

$\square$

## C  Newsvendor Problem - Additional Details

We provide additional details about our Newsvendor experiment in Section 3 when $\mathbb{P}^\star$ is a Gaussian distribution with $\mu_\star = 25$ and $\sigma_\star^2 = 100$.

**Hyperparameters.**  The prior and posterior are normal-gamma distributions. We set the prior hyperparameters to be $\mu_0 = 0$ and $\kappa_0, \alpha_0, \beta_0 = 1$. The derivation of the hyperparameters can be found in Definition 2.

**Values of $\epsilon_{\min}$ and $\epsilon^\star$.**  From Table 1 and equation (7), the value of $\epsilon_{\min}$ is 0.047. From equation (9), the average value of $\epsilon^\star$ over all $J$ seeds is 0.089 with standard deviation 0.048.

**Implementation.**  We implemented the dual problems for DRO-BAS (Theorem 1) and BDRO (Shapiro et al., 2023) in Python using CVXPY version 1.5.2 and the MOSEK solver version 10.1.28. Our implementation uses disciplined parametrized programming (Agrawal et al., 2019) which - after an initial warm start for seed $j = 1$ - allows us to solve subsequent seeds $j = 2, \dots, J$ rapidly (see Table 2). We used a 12-core Dual Intel Xeon E5-2643 v3 @ 3.4 Ghz with 128GB RAM.

**Out-of-sample mean and variance.**  For a given $\epsilon$ and seed $j$, let the optimal solution be $x^{(j)}(\epsilon)$. We calculate the out-of-sample mean $m^{(j)}(\epsilon) = \mathbb{E}_{\xi\sim\hat{\mathbb{P}}_m^{(j)}}[f(x^{(j)}(\epsilon),\xi)]$ and variance $v^{(j)}(\epsilon) = \mathrm{Var}_{\xi\sim\hat{\mathbb{P}}_m^{(j)}}[f(x^{(j)}(\epsilon),\xi)]$ of the objective under the empirical test distribution $\hat{\mathbb{P}}_m^{(j)}$. For a given $\epsilon$, the out-of-sample mean $m(\epsilon)$ and variance $v(\epsilon)$ across all seeds is

$$m(\epsilon) = \frac{1}{J}\sum_{j=1}^J m^{(j)}(\epsilon) \qquad v(\epsilon) = \frac{1}{J}\sum_{j=1}^J v^{(j)}(\epsilon) + \frac{1}{J-1}\sum_{j=1}^J \left(m^{(j)}(\epsilon) - m(\epsilon)\right)^2.$$

The out-of-sample variance $v(\epsilon)$ is equal to the mean of the variances $v^{(j)}(\epsilon)$ plus the variance of the means $m^{(j)}(\epsilon)$ (Gotoh et al., 2021).

**Solve time.**  On the initial warm-start seed $j = 1$, for each $N$, DRO-BAS solves the dual problem from Theorem 1 faster than the BDRO dual problem. For example, when $N = 900$, DRO-BAS solves problems in 0.27 seconds on average, whilst BDRO solves problem in 5.56 seconds. These results suggest that, for fixed $N$, if the solve is started from scratch with no warm start, then DRO-BAS will solve instances faster than BDRO. For seeds $j = 2, \dots, J$, disciplined

Table 2: Average (AVG) and standard deviation (STD) of the solve time for initial warm start on seed $j = 1$ and for subsequent seeds $j = 2 \ldots, J$. Distribution $\mathbb{P}^\star$ is a Gaussian $\mathcal{N}(25, 100)$.

| | Solve time in seconds AVG (STD) | | | |
| | $j = 1$ | | $j = 2, \ldots, J$ | |
| $N$ | DRO-BAS | BDRO | DRO-BAS | BDRO |
|---|---|---|---|---|
| 25 | 0.02 (0.00) | 0.07 (0.02) | 0.01 (0.00) | 0.01 (0.00) |
| 100 | 0.03 (0.00) | 0.15 (0.03) | 0.02 (0.00) | 0.03 (0.00) |
| 900 | 0.27 (0.02) | 5.56 (0.05) | 0.40 (0.11) | 0.23 (0.02) |

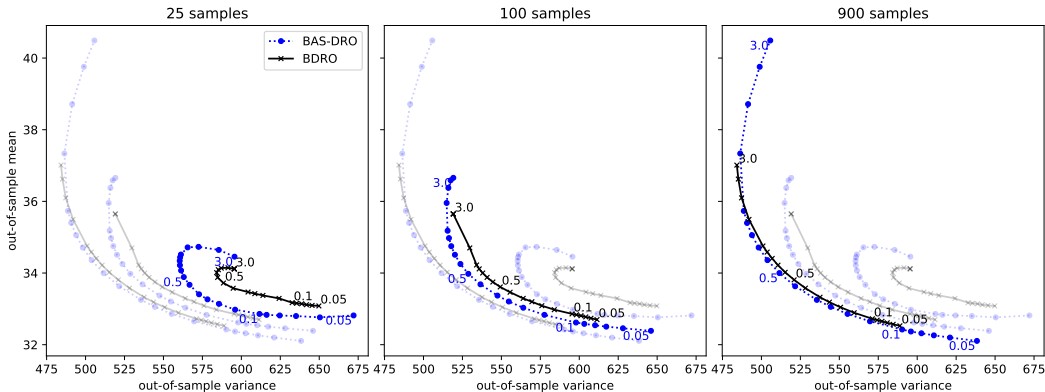

Figure 3: The out-of-sample mean-variance tradeoff on the *truncated-normal* dataset when varying the radius $\epsilon$ for DRO-BAS and BDRO when the total number of samples from the model is 25 (left), 100 (middle), and 900 (right).

parametrized programming (DPP) significantly speeds up the solve for BDRO: the average solve time for seeds $j = 2, \ldots, J$ is 0.40 seconds when $N = 900$. In contrast, DPP does not speed up the solve for DRO-BAS: the average solve time for seeds $j = 2, \ldots, J$ is 0.40 when $N = 900$. We conjecture that the speed up for BDRO using DPP is because BDRO has $N_\theta$ Lagrangian dual variables compared to DRO-BAS having exactly one Lagrangian dual variable. BDRO then benefits from the warm start because it can reuse the presolve effort spent on the $N_\theta$ dual variables spent during the warm start.

## D  Misspecified Supplementary Experiments - Truncated Normal

In this section, we present additional experiments when the data-generating process $\mathbb{P}^\star$ is a truncated normal distribution. The truncated normal has mean $\mu_\star = 10$ and variance $\sigma_\star^2 = 100$. The likelihood is a Gaussian distribution, so our model is misspecified. The conjugate prior and posterior are still normal-gamma distributions with the same hyperparameters as Section 3. The experimental setup is also the same as Section 3: the values of $\epsilon$, $N$, $n$, $m$, and $J$ are all specified the same.

**Analysis.**  When the likelihood is misspecified, Figure 3 shows the out-of-sample mean-variance tradeoff is again a Pareto front. This is the same conclusion as the well-specified case in Section 3. Furthermore, when $N = 900$, DRO-BAS has a small advantage on the mean-variance tradeoff.

**Solve time.**  Table 3 shows the same conclusions about the solve time from Appendix C can be made about the solve time for the truncated normal data-generating process.

Table 3: Average (AVG) and standard deviation (STD) of the solve time for initial warm start on seed $j = 1$ and for subsequent seeds $j = 2 \ldots, J$. Distribution $\mathbb{P}^\star$ is a truncated normal.

| | Solve time in seconds AVG (STD) | | | |
| | $j = 1$ | | $j = 2, \ldots, J$ | |
| $N$ | DRO-BAS | BDRO | DRO-BAS | BDRO |
|---|---|---|---|---|
| 25 | 0.03 (0.00) | 0.07 (0.02) | 0.01 (0.00) | 0.01 (0.00) |
| 100 | 0.03 (0.00) | 0.14 (0.02) | 0.02 (0.01) | 0.03 (0.00) |
| 900 | 0.27 (0.02) | 5.54 (0.04) | 0.40 (0.11) | 0.23 (0.01) |

