# OpenReview forum: "Distributionally Robust Optimisation with Bayesian Ambiguity Sets"
_NeurIPS.cc/2024/Workshop/BDU — NeurIPS BDU Workshop 2024 Oral_

### Official Review · Reviewer_kGU1 · 2024-09-27

**Rating:** 8
**Confidence:** 3

**Review:**

This paper introduces a new method, DRO-BAS, designed to improve decision-making under uncertainty by mitigating the risks associated with noisy data and model misspecification. By extending traditional Bayesian DRO approaches, the authors propose optimising the worst-case risk using ambiguity sets informed by posterior distributions. They demonstrate the effectiveness of DRO-BAS through theoretical proofs and a numerical example using the Newsvendor problem.

Pros:
- DRO-BAS showcases better out-of-sample performance than existing Bayesian DRO (BDRO) techniques, particularly in scenarios with small sample sizes. Figure 2 clearly illustrates that DRO-BAS dominates BDRO in mean-variance tradeoff when data is limited.
- The paper provides a clear theoretical foundation, including a closed-form dual representation for a wide range of exponential family models.
- The DRO-BAS method simplifies the dual formulation into a single-stage stochastic program, reducing computational complexity.

Cons:
- While the DRO-BAS approach works well for exponential family models, its extension to more general, high-dimensional problems is only briefly discussed as future work. This limits its immediate applicability to broader real-world decision-making problems.
- While DRO-BAS outperforms BDRO for small sample sizes, the performance gap diminishes as the number of samples increases. For large datasets, both methods show similar out-of-sample performance.
- The method introduces additional complexity in selecting the ambiguity set's tolerance level.

---

### Official Review · Reviewer_VXbq · 2024-10-03
**Review of DRO-BAS**

**Rating:** 6
**Confidence:** 3

**Review:**

The paper introduces a novel objective function for decision-making under uncertainty, which is a worst-case expected loss. The worst case is determined from a set of distributions that, a posteriori, are expected to belong to an ambiguity set of the data-generating process. The authors present an upper bound for this objective function and, for some exponential family models, derive a closed-form expression. They also empirically compare the out-of-sample mean-variance trade-off of their approach with the Bayesian DRO method described in Shapiro (2023).

I found the paper both interesting and well-written, with a potentially promising idea. However, based on the presented results, I'm unsure of the concrete advantages it offers over Bayesian DRO. The advantage seems limited to cases where a small sample is drawn from the posterior and likelihood in Bayesian DRO, and the proposed objective function has a closed-form expression. That said, when this closed-form expression holds, it doesn't seem particularly computationally demanding to draw a larger sample from the posterior and likelihood, allowing Bayesian DRO optimization to achieve the same Pareto front.

---

### Official Review · Reviewer_ov2D · 2024-10-06
**Very interesting theoretically sound approach for robust optimization**

**Rating:** 9
**Confidence:** 3

**Review:**

# Summary

This is an excellent work providing a new formulation for distributionally robust optimization in a principled manner via Bayesian ambiguity sets. The work improves over the previous related literature by providing a more suitable/principled objective (Eq. 3), which truly expresses the ambiguity set based on the current Bayesian posterior (effectively, using the correct ordering for where the expectation is taken). Then it provides solutions for some cases for the KL divergence and members of the exponential family that satisfy condition in Eq. 5. Finally, it empirically shows the method on one benchmark problem, the Newsvendor Problem.

# Comments

This paper is top-notch. Despite my lack of familiarity with the specific area, it was easy to follow along and I find the rationale and execution very compelling. Robust optimization and decision making (which naturally includes robust estimation) under posterior indeterminacy (which could be due to model misspecification) is a research area of major interest and this paper provides a significant and original contribution, to my knowledge. I think it's an excellent contribution and of broad interest especially for this venue.

In terms of limitations, the obvious ones would be addressing a broader class of distributions or providing tools or suggestions for solving Eq. 3 in more general cases. The authors might want to discuss the possible limitations of using the KL as a divergence, and possibilities of using other divergences (e.g., it's not clear if that would be possible; the current results seem to rely heavily on various KL properties).

---

### Decision · Program_Chairs · 2024-10-09

Accept (Oral)